# Conservative Therapies for TMJ Closed Lock: A Randomized Controlled Trial

**DOI:** 10.3390/jcm11237037

**Published:** 2022-11-28

**Authors:** Paola Di Giacomo, Carlo Di Paolo, Erda Qorri, Roberto Gatto, Giovanni Manes Gravina, Giovanni Falisi

**Affiliations:** 1Department of Oral and Maxillo-Facial Sciences, Sapienza University of Rome, 00185 Rome, Italy; 2Department of Dentistry, Faculty of Medical Sciences, Albanian University, 1001 Tirana, Albania; 3Department of Life Health and Environmental Sciences, University of L’Aquila, 67100 L’Aquila, Italy

**Keywords:** occlusal splints, ADDWoR, temporomandibular disorders, arthralgia, headache

## Abstract

Background. Acute anterior disc displacement without reduction (ADDWoR) is characterized by permanent TMJ disc displacement, pain and functional limitations. Occlusal appliances (OA) are among the therapies of choice. Methods. A single-blind randomized study was carried out to compare the therapeutic success of two different types of splints in patients with ADDWoR. A total of 30 subjects were eligible for the study out of the 330 screened. Group I (n = 15) received RA.DI.CA splint therapy and Group II (*n* = 15) received stabilization splint therapy. Temporomandibular pain, headache, neck pain and functional excursions were evaluated at baseline (T0), after 4 weeks (T1) and after 6 months (T2). Descriptive and inferential statistics were performed. Results. There was a significant increase in maximum jaw opening and a reduction in pain in both groups (*p* < 0.05), except for neck pain in Group II. Significant differences in between- and within-subject factors emerged in all of the parameters evaluated, especially between T1 and T2 scores, with a greater trend of improvement in Group I than Group II. Conclusion. RA.DI.CA splints were found to be more effective for the considered sample, especially in the treatment of comorbidities and functional movements, probably due to the greater orthopedic action and joint mobilization.

## 1. Introduction

Anterior disc displacement without reduction (ADDWoR) is an intra-capsular joint disorder affecting the temporomandibular joint, characterized by the alteration of the functional relationship between the mandibular and articular discs, as reported in the Diagnostic Criteria for Temporomandibular Disorders (DC/TMD) [1,2]. The acute phase is characterized by restrictions in jaw movement, due to the permanent anterior disc displacement in relation to the condylar position. Furthermore, it is associated with pain during dynamic and/or spontaneous tasks, both in the affected joint and, less frequently, in the contralateral one [2,3].

Most patients seek treatment when pain and jaw limitations negatively impact daily activities. Therefore, the reduction in or elimination of pain and functional restrictions are fundamental parameters in the evaluation of a therapeutic approach. Treatment is recommended since the pain is not self-resolving [4]. Several therapeutic approaches have been proposed in the scientific literature, classified as conservative and surgical, and are supported by pharmacological and physical therapies [5,6,7,8]. Conservative therapies continue to be the most effective way to manage 90% of patients with TMD, and surgery is the last resort if the previous therapies are not resolutive [9]. Occlusal appliances represent the most common therapy, even if there are no guidelines about a specific splint to be used. The suggested function of occlusal appliances is a decrease in painful symptomatology through the decompression of joint structures and muscle relaxation, not accounting for the placebo effect [10,11]. However, there are no studies that report the stability of these effects over time, nor guidelines stating which type of splint is actually more effective in the treatment of ADDWoR. In fact, the literature is full of articles concerning the most frequent intracapsular disorders, such as disc dislocation with reduction, and oriented towards the use of anterior-repositioning splints as a therapeutic solution. However, as far as disc dislocation without reduction is concerned, guidelines have not yet been defined to orientate therapy more specifically. In support of this, we cite the fact that there is no unanimity on the possibility of recapturing the articular disc with the occlusal splint, and consequently clinicians will choose a different therapeutic option depending on whether or not they endorse this option.

Among the therapeutic alternatives proposed in the literature, there are two groups of splints: stabilization splints and distraction ones.

The stabilization splint seems to improve muscle relaxation and reduce stress on joint structures [12], acting mainly in a static position and without searching for disc recapture, with greater effectiveness in TMD patients compared to TMD-free groups [13]. Other authors report that the use of stabilization splints for arthralgia is sufficiently suggested by the literature [14].

In the group of distraction splints, the authors of the study report their patented splint called the RA.DI.CA. splint (acronym for Rampello–Di Paolo-Cascone License No. 91-000571, 3 September 1991), designed with the intention to reduce painful symptoms and recover mandibular functionality through a dynamic orthopedic rehabilitation, consisting of joint mobilization. As reported in previous studies, the authors have established a specific protocol for ADDwoR treatment accompanied by pharmacological and physical therapies, within the limits of and in accordance with the individual characteristics of the patients [15]. Therapeutic effectiveness has already been proved; however, comparisons with other kinds of splints have not been reported. Therefore, the authors aimed to compare the effectiveness of the RA.DI.CA and the stabilization splints in subjects affected by acute ADDwoR. The hypothesis is to assess whether a device with a more dynamic action yields different results compared to a device with more static characteristics. The RA.DI.CA splint was used as the treatment and the stabilization splint as the active control.

## 2. Materials and Methods

In order to address the aim of this research, the authors developed a single-blind randomized study, with two treatment arms:RA.DI.CA splint (Group I) and stabilization splint (Group II).

The study was conducted between July 2020 and May 2022 at the Gnathology Service of the Integrated Head and Neck Care Department of Policlinico Umberto I, “Sapienza” University of Rome.

The study was approved by the Institutional Ethics Committee of Sapienza University (no. 349 24 October 2018) and by the Institutional Ethics Committee of Albanian University (Approval No. 139—22 February 2022). The study has also been registered in the ISRCTN registry with the number protocol 77777174.

The research was conducted in accordance with the World Medical Association’s Code of Ethics (Declaration of Helsinki) for experiments on humans.

### 2.1. Participants and Sampling

**Sample calculation** was made on the basis of an estimate of subjects with ADDwoR who visited the Department of Oral and Maxillo-Facial Sciences in the previous years.

Each year, approximately 350 subjects with temporomandibular disorders are screened for the first time, and according to the epidemiological prevalence data per year, at our department as well, about 5% (18 subjects) were affected by ADDwoR.

Sample size was calculated by applying the correction for a small population, with a confidence level of 95%, a confidence interval of 5% and a power of 80%.

The estimated number of subjects was 24.

Subjects who came to be examined at the Department of Oral and Maxillo-Facial Sciences, Gnathology Service—Policlinico Umberto I, Sapienza University of Rome were screened by expert and calibrated clinicians, according to the DC/TMD Criteria, within a period of 15 months.

***Inclusion and exclusion criteria*** for sampling are reported below.

Inclusion criteria: (a) Patients over 18 years old; (b) diagnosis of jaw functional limitation due to unilateral disc displacement without reduction for less than 6 months, which was verified in MRI (Magnetic Resonance Imaging) according to the RDC/TMD (Research Diagnostic Criteria for Temporomandibular Disorders) and DC/TMD criteria.

Exclusion criteria: (a) Jaw limitation due to other factors such as connective tissue diseases (sclerodermia), traumas (mandibular or condylar fractures), deformities, tumors, TMJ ankylosis or muscular locking; (b) other concomitant conservative therapies such as physical rehabilitation.

**Diagnosis of ADDwoR** was made by collecting signs and symptoms suggestive of this pathology, such as the deviation of the mandible towards the affected side during opening movement, a negative end-feel test, limitations of the mouth opening and pain in the affected joint and less frequently in the contralateral one.

MRI was performed only with the aim of confirming the differential diagnosis with other pathologies with overlapping signs and symptoms.

### 2.2. Study Design

***Randomization.*** Subjects eligible for the study were randomly divided into two groups by a provider not involved in the initial screening and clinical assessments, using a computer-generated blocked random allocation sequence with a block size of 2. Participants were told that they had an equal chance of being assigned to one of the two treatments, as described in the informed consent.

Another provider, blinded to the type of treatment the patient had undergone, was responsible for data collection at the end of the treatment.

Group I (*n* = 15) received RA.DI.CA splint therapy and Group II (*n* = 15) received stabilization splint therapy.

**Study phases.** The study included three phases:

**T0** = Start of therapy;

**T1** = 4 weeks from the start of therapy;

**T2** = Follow-up after 6 months of therapy.

***Variables of the study are*** (***a***) Sociodemographic factors such as gender, age, marital status and occupation. (***b***) Occlusions such as occlusal and skeletal class, dental formula, occlusal abnormalities, incisal guide, loss of teeth and parafunctions. These parameters were collected on the basis of clinical examination by using standard lateral radiographs and dental orthopantomography. (***c***) Medical history of patients, including previous trauma, previous click and beginning of symptoms. (***d***) Types of pain, including temporomandibular pain (arthralgia/muscle pain), headache, familiar pain, neck pain and emotional strain. According to the verbal numeric scale (VNS), reporting values range from 0 = no pain to 100 = severe pain perceived by the patient at the baseline (T0), after 4 weeks (T1) and 6 months (T2) from the beginning of therapy. These values can be divided into four categories: mild (0–20); moderate (20–50); strong (50–80); and severe (80–100). (***e***) Functional aspects: maximum mouth opening with a normal value of 45 ± 5 mm [5] and lateral excursions, expressed in mm at the baseline (T0), after 4-week therapy (T1) and after 6-month therapy (T2) by means of a gauge. (**f**) Number of days necessary for the resolution of closed-lock and the degree of resolution. (**g**) Presence of a click after the treatment, and perception of any occlusal change.

**Treatments and protocols.***Stabilization splints* were made using autopolymerizing acrylic resin with bilateral occlusal contacts on the flat surface (Figure 1), checked in the articulator, according to Ash’s recommendations [16]. A full-arch splint was used, designed using the model of the upper arch with retention hooks at the premolar level. The clinician must ensure that the device fits correctly. The occlusal contacts were checked intraorally for balance. The action performed by this type of splint is promoting muscle relaxation and decompression of joint structures by increasing the vertical dimension in a static position [12].

A *RA.DI.CA.* splint (Figure 2) is composed of:A heat-cured acrylic resin upper plate (**1**),A heat-cured acrylic resin lower plate (**2**),An anterior hinge (**3**),Two vestibular springs made with orthodontic wire (**4**),Two or more Adams clasps and/or two ball clasps (**5**),A vestibular steel arch (**6**).

The upper plate has two surfaces. The inner one, also called the “occlusal/palatal” surface, adapts to the masticatory surfaces of the upper teeth and to the anterior 2/3 of the palatal vault. The outer surface is smooth and faces the lower plate. Two Adams clasps are placed on the right and left first molars and the ball clasps are placed in the interdental space of the right and left first and second premolars. The upper plate plays a stabilizing role, having a double mucous and a dental anchorage.

The lower plate, also shaped like a horseshoe, has two smooth surfaces. One is in contact with the outer surface of the upper plate and the other with the masticatory surfaces of the lower teeth. The latter should be shaped in accordance with Spee and Wilson curves.

The two plates are connected to each other by a front hinge placed at the incisor level and by two (left and right) vestibular springs connected in the canine–premolar site, in order to give an elastic resistance to the lower plate during the functional movements of mouth closure. The springs can be hard or soft according to the characteristics of the patient.

A RA.DI.CA. splint is a dynamic occlusal appliance that performs active joint mobilization during the closing movement of the mouth, unlike other distraction splints. However, another passive distraction of the visco-elastic joint and muscle components is realized during the opening movement of the mouth. The functional action is carried out through a “pushing” mechanism at the level of the posterior area of the lower dental arch, which induces a downward and forward movement of the mandibular condyle. Therefore, by means of its components, the illustrated device has as its main therapeutic goal the recovery of the functional relationship between the articular disc and the mandibular condyle, and, in more severe cases where recovery is not achievable, the rehabilitation of jaw function and the reduction in -algic symptomatology. In these cases, it is possible to take advantage of joint mobilization, as it is in the orthopedic treatment of other joints of our body, which is also reported in the study of Nitzan et al. [17]. Furthermore, joint mobilization aims to speed up the fibrosis of a part of the posterior bilaminar area with consequent reduction in pain [18].

The patients in both treatment groups were instructed to wear the splint for 2 h during the daytime and at night. At each follow-up visit, occlusal contacts were checked, and they were corrected where necessary. No additional drugs or physiotherapy sessions were provided for either of the two groups.

### 2.3. Statistical Analysis

Descriptive statistical analysis (percentage, average, median, mode, standard deviation and minimum and maximum values) was performed for each variable of the study, collected at the baseline (T0), after 4 weeks (T1), and after six months from the start of the therapy (T2). A Student’s *t*-test was used to assess statistical significance between groups for the baseline symptomatology scores with *p* < 0.05.

A generalized linear mixed model (GLMM) with a chi-square test for within- (effect of time and the interaction between treatment and time) and between-subject factors (effect of type of treatment) was performed to allow both fixed and random effects and was particularly used when there was non-independence in the data. The GLMM describes the relationship between a response variable (i.e., pain scores) and other explanatory variables such as fixed ones (treatment and time) and random ones. An ANOVA summary was reported.

An analysis of covariance—ANCOVA—was performed in order to estimate a correlation between baseline score and follow-up measures for each treatment, and the mean differences between treatments.

Post hoc tests were performed. A two-tailed value of *p* < 0.05 was regarded as significant. All analyses were performed with SPSS version 27.0.

## 3. Results

The flow of patients through the study, according to CONSORT criteria, is reported below (Figure 3). Five hundred and eighty subjects were screened over a period of 15 months, and five hundred and fifty were excluded because they did not meet the inclusion criteria and/or met one or more exclusion criteria.

### 3.1. Descriptive Statistics

The sample was composed of 30 subjects:24 females (80%) and 6 males (20%), with a mean age of 39.25 ± 12.70. In total, 19 subjects (63%) had Class I dental occlusions and 11 subjects (37%) had Class II dental occlusions. Further, 11 subjects (37%) had a normal incisal guide, 3 subjects (10%) had a deep incisal guide and 16 subjects (53%) had a vertical incisal guide.

The average onset of symptoms (days) was 55 ± 15.99. All patients referred to previous joint noises, 5 subjects (17%) previous traumas and 18 subjects (60%) parafunctions.

At baseline, according to the VNS Scale, strong pain with values ranging between 50 and 80 was found in all of the parameters analyzed (arthralgia, headache and neck pain) and mouth opening had an average value below the normal range of 45 ± 5 mm [5]. Statistically significant differences between groups were not found in baseline pain and functional assessments, as reported in the Student’s *t*-test with *p* < 0.05 (Table 1).

### 3.2. Inferential Statistics

There was a significant increase in maximum jaw opening and reduction in pain in both groups (*p* < 0.05), as reported in the post hoc test, except for neck pain in Group II (*p* Tukey = 0.665), as shown in Table 2.

As for between-subject factors, there is an effect of treatment with a statistical significance of *p* < 0.05 in all of the parameters analyzed (arthralgia, headache, neck pain and mouth opening), in particular between T0 and T2 and T1 and T2 scores (Table 3). The greatest mean difference between VNS scores is found in the headache parameter (Table 4).

As for within-subject factors, there is an effect of time with a statistical significance of *p* < 0.05 in all of the parameters analyzed and time lapses. There is also a recognizable and significative interaction between treatment and time, except for jaw opening between T0 and T1 (Table 3).

Figure 4 shows the trend in symptomatological and functional values in relation to time and type of treatment used.

In patients using the stabilization splint, there were no statistically significant differences between T0 and T2 for neck pain VNS scores, determined by comparing the values in Table 1 and Table 2 (T0 = 53 ± 20.02 and T2 = 42.667 5.080).

Baseline scores are predictors for follow-up scores as reported in the ANCOVA test (Table 3). In Figure 5, the independent variable (on the *x*-axis), which is the baseline value, has proven to predict the value of the dependent variable (on the *y*-axis), which is the follow-up score.

## 4. Discussion

In the scientific community, surgical and non-surgical options are proposed for the treatment of ADDwoR. Among the non-surgical options, occlusal splints are the most widely used [5,6,7,8,9]. There is a huge variety of opinions about a splint’s specific action and design, and many of them share common actions. There are three types of occlusal splints that are commonly used in patients with jaw function limitations: the stabilization, the distraction/pivot and the anterior repositioning splints [19,20]. As reported in the studies of Schmitter and Stiesch-Scholz [6,10], both functional and symptomatic recovery after the analyzed therapies seemed to be superimposable. However, the comparison of the results from various studies was made difficult by the discrepancies in evaluating the success of a treatment method.

The effectiveness of the RA.DI.CA splint was evaluated in previous studies [5,15]; however, in this research, the authors aimed to compare it to the stabilization splint in order to identify advantages and limits of each therapy for the sake of assessing the most specific and effective therapeutic protocol for this kind of pathology. The authors chose the stabilization splint because of its wide use and easy fabrication and management for the treatment of temporomandibular joint disorders and, among them, ADDwoR. However, there is no agreement in the scientific literature about the stabilization splint’s therapeutic effects. As reported in the introduction section, the stabilization splint seems to provide several positive effects on function and symptoms. However, when compared with other therapies or control groups, no statistical significance has emerged from its therapeutic effect. Lund et al. reported that the therapeutic impact of stabilization splints was found in 32% of patients [10], and that there was no significant difference between the outcomes of stabilization splint therapy and the control group. In the comparison between the stabilization splint and the pivot splint (distraction-type splint), other authors [6,10] reported that the success of the therapy in both groups of patients was not dependent on the type of occlusal splint in the two treatment groups, and no significant discrepancy was found.

The results of the current study showed an improvement in both treatment groups, especially in arthralgia, followed by headache and mouth opening with *p* < 0.05, with an effect of patient baseline score as a predictor of follow-up scores. However, the comparison of the two treatments showed a statistically significant greater improvement in patients using a RA.DI.CA splint in all of the parameters analyzed (*p* < 0.001). Furthermore, RA.DI.CA splint therapy improved neck pain (*p* < 0.05), which remained unchanged in the stabilization group (*p* = 0.01). (Table 2). The magnitude of the improvement in joint function and symptomatology could be superimposed on what has been found in other studies with other distraction-type splints [21]. No other study has reported assessments of other comorbidities, such as headache and neck pain.

Compared to the study of Schmitter and Stiesch-Scholz [6,10], the greater effectiveness of RA.DI.CA splints is due to the greater amount of distraction and orthopedic action, allowed by its design and the type of springs used. Compared to the previously mentioned one, a pivot splint [22] acts at the level of the second molar during the opening movement of the mandible so that it can turn in the anterior cranial direction, producing a caudal movement of the condyle, as described by Sears [23]. However, distraction strength is not standardized and measurable because it is managed by the patient. RA.DI.CA. splints act during closing movements by the induction of a downward and forward movement of the mandibular condyle with a strength standardized and coded by the springs used. This way, the splint “increases” the intra-articular virtual spaces, blood flow and tissue tropism and reduces the load on the retrodiscal (bilaminar) structures [15]. The device aims to reproduce the “Martini maneuver”, which proved to be effective in unlocking subjects with ADDWoR [6,24]. RA.DI.CA splints act as an “intra-oral physiotherapy” and allow an active and passive gymnastic with a gradual action in order to recover not only articular functionality, but also muscular. This seems to also be more effective in comorbidities such as neck pain. The limitations of the study are (a) the small sample size and (b) no systematic follow-up data beyond six months. The small sample size is due to the low prevalence of ADDWoR (5%) [1]. In the population examined, this percentage is superimposable. However, although the clinicians screened a medium–large number of subjects, only a small percentage were enrolled. Despite this, the sample size obtained at the end of the screening process was adequate to enable the comparison between the averages of the two groups, with a power of 80% and a confidence level of 95%.

## 5. Conclusions

The importance of treating this type of pathology, even though it is present in only 5% of the population, is linked to the invalidating aspects in daily life referred to by the patients, as well as the possibility of preventing more severe disorders that would otherwise only have a surgical resolution. The difference in results between the two splints is probably due to the greater orthopedic action of RA.DI.CA splints, with a higher effectiveness in the degree of joint mobilization and a positive repercussion on symptomatology such as headache and neck pain. It should also be noted that the aim of these therapies is not the recovery of the disc position, but joint mobilization and the achievement of a functional condyle–disc relationship with a better joint load distribution that avoids the progression of structural damage and the recurrence of symptoms.

## Figures and Tables

**Figure 1 jcm-11-07037-f001:**
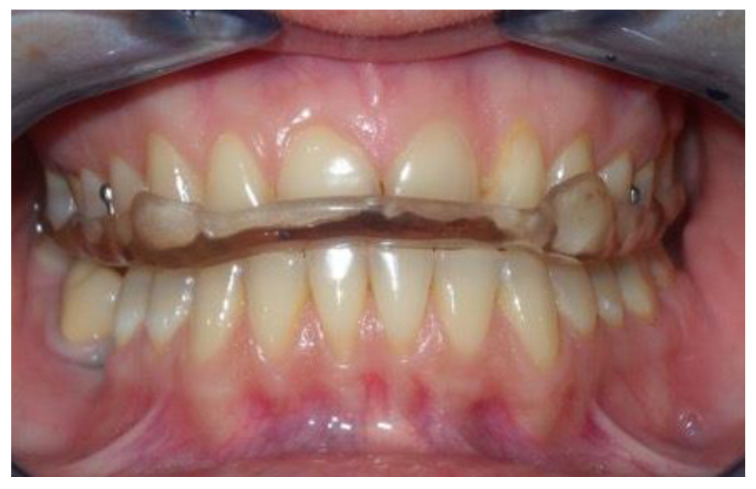
Stabilization splint.

**Figure 2 jcm-11-07037-f002:**
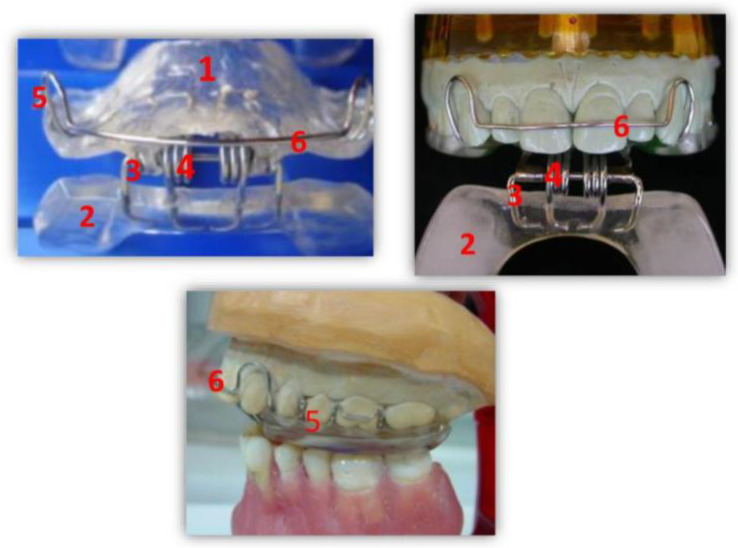
RA.DI.CA. splint and its components.

**Figure 3 jcm-11-07037-f003:**
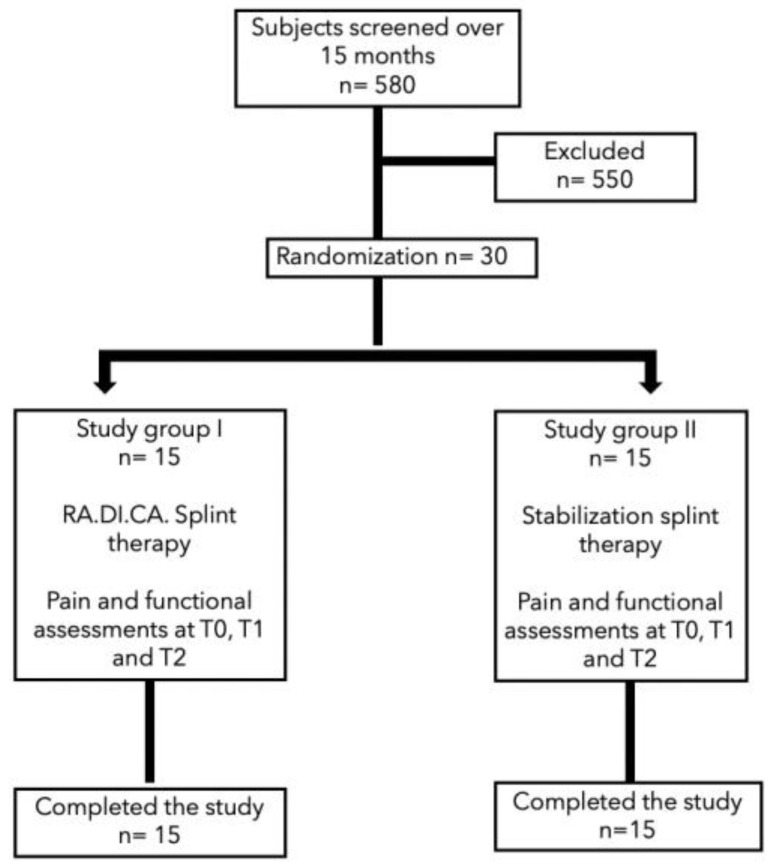
CONSORT criteria.

**Figure 4 jcm-11-07037-f004:**
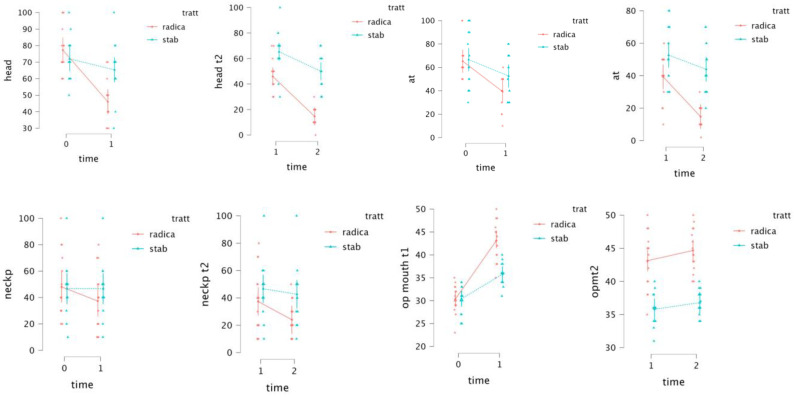
Plots from the GLMM.

**Figure 5 jcm-11-07037-f005:**
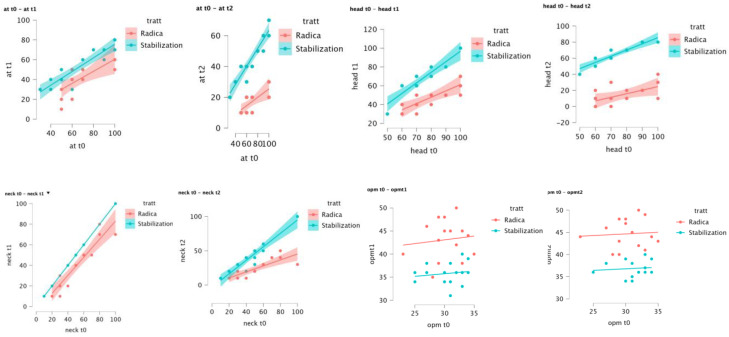
Plots from ANCOVA.

**Table 1 jcm-11-07037-t001:** Baseline general and clinical characteristics for each study group. Pain scores are reported according to verbal numeric scale values and mouth opening is expressed in mm. Student’s *t*-test with *p* < 0.05.

Parameter Assessed	Stabilization Splint		RA.DI.CA Splint		*p*-Value Student’s
	Mean	SD	Mean	SD	
**Arthralgia**	68.00	24.85	69.00	17.28	0.89
**Headache**	75.00	13.54	83.00	13.37	0.11
**Neck pain**	53.00	20.02	57.00	24.51	0.62
**Mouth opening**	31.30 (mm)	2.66	30.40 (mm)	3.47	0.43

**Table 2 jcm-11-07037-t002:** Estimated Marginal Means (EMM) for each treatment and time (T1 and T2) with 95% Confidence Interval (CI).

Parameter Assessed at T1	RADICA EMM ± SD		95% CI for Mean Difference		Stabilization Splint Mean ± SD		95% CI for Mean Difference	
			Lower	Upper			Lower	Upper
Arthralgia	39.333	4.842	29.843	48.823	52.667	4.842	43.177	62.157
Headache	46.000	3.695	38.759	53.241	65.333	3.695	58.092	72.575
Neck pain	37.333	5.711	26.139	48.527	46.667	5.711	35.473	57.861
Mouth opening	43.133	0.842	41.484	44.783	35.800	0.842	34.150	37.450
Parameter assessed at T2								
Arthralgia	14.800	3.727	7.496	22.104	44.000	3.727	36.696	51.304
Headache	14.667	3.469	7.868	21.465	50.000	3.469	43.202	56.798
Neck pain	24.000	x5.080	14.043	33.957	42.667	5.080	32.709	52.624
Mouth opening	44.667	0.791	43.116	46.217	36.800	0.791	35.249	38.351

**Table 3 jcm-11-07037-t003:** Generalized linear mixed model (GLMM) with ANOVA summary for within- and between-subject factors, with statistical significance set at *p* < 0.05.

**Parameter Assessed**	**Within-Subjects Factor (Time T0** **→T1)**		**Between-Subjects Factor (Type of Treatment)**		**Interaction between Treatment and Time**	
	**Chi-Square value**	***p* value**	**Chi-Square value**	***p* value**	**Chi-Square value**	***p* value**
**Arthralgia**	44.510	**<0.001**	1.320	0.251	8.028	**0.005**
**Headache**	54.449	**<0.001**	2.046	0.153	34.575	**<0.001**
**Neck Pain**	17.307	**<0.001**	0.267	0.605	17.307	**<0.001**
**Mouth opening**	68.096	**<0.001**	13.964	**<0.001**	18.242	**<0.001**
**Parameter assessed**	**Within-subjects factor (time T1** **→T2)**		**Between-subjects factor (type of treatment)**		**Interaction between treatment and time**	
	**Chi-Square value**	***p* value**	**Chi-Square value**	***p* value**	**Chi-Square value**	***p* value**
**Arthralgia**	45.856	**<0.001**	14.929	<0.001	18.045	**<0.001**
**Headache**	41.713	**<0.001**	28.075	<0.001	8.648	**0.003**
**Neck Pain**	14.289	**<0.001**	4.147	0.042	4.886	**0.027**
**Mouth opening**	13.391	**<0.001**	30.742	<0.001	0.773	0.379
**ANOVA Summary**	**Within subjects factor (time T0** **→T2)**		**Between** **-subjects factor (type of treatment)**		**Interaction between treatment and time**	
	F	***p* value**	**F**	***p* value**	**F**	***p* value**
**Artralgia**	170.163	**<0.001**	6.244	**0.019**	23.379	**<0.001**
**Headache**	219.299	**<0.001**	21.928	**<0.001**	131.403	**<0.001**
**Neck Pain**	23,748	**<0.001**	1660	0.208	13,206	**<0.001**
**Mouth opening**	157.609	**<0.001**	37.791	**<0.001**	24.227	**<0.001**

**Table 4 jcm-11-07037-t004:** Analysis of covariance for T0/T1 and T0/T2 with an ANCOVA test, with statistical significance set at *p* < 0.05. Mean difference between T0 and T2 with a 95% CI between RA.DI.CA and stabilization splints for each pain and functional parameter.

Parameters	T0 → T1		T0 → T2	
	F	*p*-Value	F	*p*-Value
**Arthralgia**	66.741	<0.001	87.227	<0.001
	95% CI for Mean Difference			
Mean difference between T0 and T2	Lower	Upper	ptukey	
−27.337	−31.728	22.946	<0.001	
**Headache**	54.712	<0.001	289.605	<0.001
	95% CI for Mean Difference			
Mean difference between T0 and T2	Lower	Upper	ptukey	
−52.500	−58.830	−46.170	<0.001	
**Neck pain**	22.164	<0.001	66.982	<0.001
	95% CI for Mean Difference			
Mean difference between T0 and T2	Lower	Upper	ptukey	
−20.876	−28.041	−13.711	<0.001	
**Mouth opening**	0.419	0.523	0.190	<0.666
	95% CI for Mean Difference			
Mean difference between T0 and T2	Lower	Upper	ptukey	
7.871	5.871	9.871	<0.001	

## Data Availability

The authors confirm that the data supporting the findings of the study are available within the article and the datasets used and/or analyzed during the current study are available from the corresponding author on reasonable request.

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
