# Peer review of "Conservative Therapies for TMJ Closed Lock: A Randomized Controlled Trial"

_jcm, 2022, doi:10.3390/jcm11237037_

Round 1
Reviewer 1 Report
The topic of the research is original and very interesting because there are not many papers published in last few years with ADDWoR problematics and OA therapy.
Author Response
Thank you for Your revision.
Reviewer 2 Report
Thank you for that manuscript. In my opinion it is a good subject of a study, to compare two different types of splints.
Abstract
- is clear and well structured.
Introduction:
- A sound and valid nullhypothesis seemed be missing.
Materials and methods:
- I think this part is lacking and you have to add informations to make the assessment of your datas more understandable.
- Sample calculations seems to be too long. In my opinion you can delete the formula.
- There seems to be something missing in the study design. In the results part (Table 2) you show the parameters within subject factors and between. Furthermore, you show the interaction between treatment and timse. You
- You should describe the differences between both splint types in more detail.
Results:
- Your sample size seems to be very small. Please discuss this in the discussion part, aswell as the short study period and the sex.
- Describe Table 1 in detail. What is the meaning of the clinical charcteristics and which values have you shown in this table accoring to the assessed parameters? Are there norm values? Please describe this in material and methods. The values in this table are not clear defined.
- Make sure, that the disclosures of p<0,05 are consistent.
- The same with Table 2: Describe the parameters within and between subjects factor and time and treatment in more detail. How have you assessed the values?
- Please provide, that the data sheets are arranged more clearer.
- Describe figure 2 and 4 in more detail and add references in the text.
Discussion:
- Please check again your references and citations style. Some are not according to the Journal’s guidelines.
- For example the reference “by Sears” is not cited.
- When the mouth opening is getting better with the therapy of RA.DI.CA, does this have an effect on the functional condyle-disc relationship?
- Add limitaions of your study in the discussion part.
Author Response
The authors would like to thank the reviewer for the proposed corrections that improved the accuracy of the manuscript.
The requested corrections have been made.
As for the statement " When the mouth opening is getting better with the therapy of RA.DI.CA, does this have an effect on the functional condyle-disc relationship?", there can be two options: the opening of the mouth improves because joint disc recapture has occurred and the second one is that there has been an action largely on the visco-elastic structures. This second eventuality occurs in those severe cases in which the dislocation is in such advance state that it cannot be reduced. Anyway, in some case it may occur a fibrosis of a part of the posterior bilaminar area with consequent reduction of pain and improving of functionality, even without disc recapturing.
Reviewer 3 Report
Thank you for the opportunity to review this manuscript. Find below my suggestions for improvement:
1. This study states: “However, there are no studies which report the stability of these effects over time, nor guidelines stating which type of splint is actually more specific in the treatment of ADDWoR.” In truth, there are many reports in the literature about the efficacy of different types of intraoral splints for the treatment of TMD intracapsular disorders. I do recommend the authors to investigate the literature deeper. Most of the studies presented in the literature review of this manuscript were collected from references with more than 10 years. If the authors search PUBMED or other scientific search engine they will find many relevant studies about differences in anterior repositioning splints used in the past. Please see some of the articles and see if they are relevant to your study (these are all old studies, but there are many new ones.) And even if the goal was not to recapture the disc, the effects on the retrodiscal tissue is similar.
a. Manzione J, Tallents RH, Katzberg RW: Arthrographically guided appliance therapy: For recapturing the temporomandibular joint meniscus. Oral Surg Oral Med Oral Path, 1984; 57:235-40
b. Tallents RH, Katzberg RW, Manzione J, Oster C, Miller T: Arthrographically assisted splint therapy. J Prosthet Dent, 1985; 53:235-8.
c. Tallents RH, Katzberg R, Miller T, Manzione JV, Oster C: Evaluation of arthrographically assisted splint therapy in treatment of TMJ disc displacement. J Prosthet Dent, 1985; 53:836-8.
d. Tallents RH, Katzberg RW, Miller TL, Manzione J, Macher DJ, Roberts CA: Arthrographically assisted splint therapy: Painful clicking with a non-reducing meniscus. Oral Surg Oral Med Oral Path,1985; 61:2-4.
e. Tallents RH, Katzberg RW, Macher DJ, Roberts CA, Sanchez-Woodworth R, Proskin HM: Use of protrusive splint therapy in anterior disk displacement of the temporomandibular joint: A 1 to 3 year follow up. J Prosthet Dent, 1990; 63:336-41.
f. Lundh H, Westesson PL, Kopp S, Tillstrom BA: Anterior repositioning splint in the treatment of temporomandibular joint with reciprocal clicking: Comparison with a flat occlusal splint and untreated control group. Oral Surg Oral Med Oral Path, 1985; 60:131-6.
g. Moloney F, Howard J: Internal derangements of the temporomandibular joint. III. Anterior repositioning splint therapy. Aust Dent J, 1986; 31:30-39.
h. Anderson GC, Schulte JK, Goodkind RJ: Comparative study of two methods for internal derangements of the temporomandibular joint. J Prosthet Dent, 1985; 53:392-7.
i. Clark GT: Treatment of jaw clicking with temporomandibular repositioning. J Craniomandib Pract, 1984; 2:263-70.
j. Okeson JP: Long-term treatment of disk-interference disorders of the temporomandibular joint with anterior repositioning occlusal splints. J Prosthet Dent, 1988; 60:611-6.
2. What were the clinical criteria to determine which patients presented with ADDWoR? If only clinical characteristics were used to determine, then these clinical signs and symptoms should be stated in the manuscript.
3. The sample calculation description is NOT necessary to be included in the study, after all, the authors must only state which statistical analysis was used to determine the sample size calculation. All the formulas on the manuscript are likely not appreciated by most of the readers, and may be seen as a distraction from the focus of the study.
4. The authors must state how they selected the participants for the study, and how they determine the groups to be ADDWoR, only by searching for clinical signs, unless everyone screened in the clinic would undergo an MRI, which then would likely be not appropriate.
5. What was the reason that this study had two IRBs, one in Italy and another in Albania?
6. We recommend the authors to name the teeth in full, and not use the international numbering system.
7. It is also important that the authors attempt to explain how the mechanics would work with the appliance used (RA.DI.CA) if the idea is NOT to attempt to recapture the disc, if the case is deemed to have the TMJ disc displace without reduction.
8. According to CONSORT Criteria, the authors again MUST explain in detail what they meant by “SCREEN FAILURE”.
9. In the discussion session, the authors claim that the RA.DI.CA appliance acts on myofibrils of actina and myosin in sarcomers, switched off by the contracture status. However, the authors have NO evidence to claim since the focus of the study was purely clinical and did not investigate tissue changes.
10. According to the small sample size of the study, would it be appropriate to call this a pilot study?
11. The authors must state in this article that they have a conflict of interest in the development of this study, since one of the authors is one of the inventors of the study.
Author Response
The authors would like to thank the reviewer for the proposed corrections that improved the accuracy of the manuscript.
The requested corrections have been made.
Reply to point 5 and 11:
The choice of having two study groups was the need to eliminate any kind of bias or interference in the interpolation of results.
Reply to point 10:
We do not consider it a pilot study as from the calculation of the sample it appears to be statistically adequate, although facing the lower limit of acceptability.
Round 2
Reviewer 3 Report
1. I would like to request the authors to revise the last sentence from the conclusion. "However, when possible, recapture of joint disc is desirable and possible with this kind of splint. This could be the basis of the stability of the RA.DI.CA. splint treatment over time." There is absolutely nothing in the study that can prove this appliance recaptures the disc. The only way this statement could be made is if authors could see inside of the joint, either by opening the joint surgically or taking an MRI. And in no part of the study those procedures were conducted at the end of the study to claim the appliance recaptures the disc. This statement can be rewritten and it shouldn't state that it is possible to recapture the disc, since there was no imaging indication of such result. The study was all based on symptoms.
2. Question # 5 from the initial review was not responded. Please address the question. "What was the reason that this study had two IRBs, one in Italy and another in Albania?" The reviewer read your explanation that you needed two study groups. However, was the sample pulled from both sites, or a combination of the sample from different countries?
3. Can the authors make a statement about the significance of the appliance to treat such a small percentage of patients who had the problem (5%)?
Author Response
- It is true. This sentence was made on the basis of our clinical practice and MRI performed during follow up in many of our patients, which allow us to verify it. However, as you affirm, MRI results are not reported in the manuscript, and we will limit to focus to the concept of dynamic function and articular mobilization. The sentence has been removed.
- The sample was collected only at Sapienza University, as reported in the materials and methods section. The reason why the authors requested the approval of the Albanian Ethics Committee is related to the analysis of medical scores in order to avoid any form of bias. In fact the analysis of data was made by clinicians who had never used RA.DI.CA. splint and who had not participated directly in the trial and thus had not influenced the results in any way.
- A sentence has been included in the conclusion, as suggested.